# Germ cell connectivity enhances cell death in response to DNA damage in the *Drosophila* testis

Kevin L Lu[1,2,3], Yukiko M Yamashita[1,2,4,5]*

[1]Life Sciences Institute, University of Michigan, Ann Arbor, United States; [2]Cellular and Molecular Biology Program, University of Michigan, Ann Arbor, United States; [3]Medical Scientist Training Program, University of Michigan, Ann Arbor, United States; [4]Department of Cell and Developmental Biology, University of Michigan, Ann Arbor, United States; [5]Howard Hughes Medical Institute, University of Michigan, Ann Arbor, United States

**Abstract** Two broadly known characteristics of germ cells in many organisms are their development as a 'cyst' of interconnected cells and their high sensitivity to DNA damage. Here we provide evidence that in the *Drosophila* testis, connectivity serves as a mechanism that confers to spermatogonia a high sensitivity to DNA damage. We show that all spermatogonia within a cyst die synchronously even when only a subset of them exhibit detectable DNA damage. Mutants of the fusome, an organelle that is known to facilitate intracyst communication, compromise synchronous spermatogonial death and reduces overall germ cell death. Our data indicate that a death-promoting signal is shared within the cyst, leading to death of the entire cyst. Taken together, we propose that intercellular connectivity supported by the fusome uniquely increases the sensitivity of the germline to DNA damage, thereby protecting the integrity of gamete genomes that are passed on to the next generation.
DOI: https://doi.org/10.7554/eLife.27960.001

*For correspondence:
yukikomy@umich.edu

## Introduction

A prevalent feature of germ cell development across species is their proliferation as an interconnected cluster of cells, widely known as a germ cell cyst. In many organisms from insects to humans, germ cells divide with incomplete cytokinesis that results in interconnected cells with shared cytoplasm, leading to cyst formation (*Greenbaum et al., 2011*; *Haglund et al., 2011*; *Pepling et al., 1999*). During oogenesis of many species from flies to mammals, this intercellular connectivity is critical for the process of oocyte specification, allowing only some of the developing germ cells to become oocytes while the others adopt a supportive role (*de Cuevas et al., 1997*; *Lei and Spradling, 2016*; *Pepling et al., 1999*). For example, in the *Drosophila* ovary, four rounds of germ cell divisions with incomplete cytokinesis results in a cyst of 16 interconnected germ cells, where only one becomes an oocyte while the remaining 15 germ cells become nurse cells. During this process, nurse cells support oocyte development by providing their cytoplasmic contents to oocytes via intercellular trafficking (*Cox and Spradling, 2003*; *de Cuevas et al., 1997*; *Huynh and St Johnston, 2004*).

In contrast to oogenesis, where cytoplasmic connectivity has a clear developmental role in oocyte development, spermatogenesis is a process where all germ cells within a cyst are considered to be equivalent and become mature gametes (*Fuller, 1993*; *Yoshida, 2016*). Despite the lack of a 'nursing mechanism' during spermatogenesis, intercellular connectivity is widely observed in spermatogenesis in a broad range of organisms (*Greenbaum et al., 2011*; *Yoshida, 2016*). While a function

for this connectivity has been proposed in post-meiotic spermatids in complementing haploid genomes (*Braun et al., 1989*), the biological significance of male germ cell connectivity during pre-meiotic stages of spermatogenesis remains unknown.

Another well-known characteristic of the germline is its extreme sensitivity to DNA damage compared to the soma, with clinical interventions such as radiation or chemotherapy often resulting in impaired fertility (*Arnon et al., 2001*; *Meistrich, 2013*; *Oakberg, 1955*). Although high DNA damage sensitivity in mammalian female may be explained by its extremely limited pool size, it remains unclear how mammalian male germline is also sensitive to DNA damage. It has been postulated that the high sensitivity of the germline to DNA damage is part of a quality control mechanism for the germ cell genome, which is passed onto the next generation (*Gunes et al., 2015*). However, the means by which the germline achieves such a high sensitivity to DNA damage remains unclear.

Here we provide evidence that germ cell connectivity serves as a mechanism to sensitize the spermatogonia (SGs) to DNA damage in the *Drosophila* testis. We show that an entire SG cyst undergoes synchronized cell death as a unit even when only a subset of SGs within the cyst exhibit detectable DNA damage. Disruption of the fusome, a germline-specific organelle that facilitates communication amongst germ cells within a cyst (*de Cuevas et al., 1997*), compromises synchronized germ cell death within a cyst in response to DNA damage. The sensitivity of a germ cell cyst to DNA damage increases as the number of interconnected germ cells within increases, demonstrating that connectivity serves as a mechanism to confer higher sensitivity to DNA damage. Taken together, we propose that germ cell cyst formation serves as a mechanism to increase the sensitivity of genome surveillance, ensuring the quality of the genome that is passed onto the next generation.

## Results

### Ionizing radiation induces spermatogonial death preferentially at the 16 cell stage

The *Drosophila* testis serves as an excellent model to study germ cell development owing to its well-defined spatiotemporal organization, with spermatogenesis proceeding from the apical tip down the length of the testis. Germline stem cells (GSCs) divide to produce gonialblasts (GBs), which undergo transit-amplifying divisions to become a cyst of 16 interconnected spermatogonia (16-SG) before entering the meiotic program as spermatocytes (*Figure 1A*). In our previous study we showed that protein starvation induces SG death, predominantly at the early stages (~4 SG stage) of SG development (*Yang and Yamashita, 2015*) (*Figure 1A*). Starvation-induced SG death, which itself is non-apoptotic (*Yacobi-Sharon et al., 2013*), is mediated by apoptosis of somatic cyst cells encapsulating the SGs (*Yang and Yamashita, 2015*). Cyst cell apoptosis breaks the 'blood-testis-barrier' and leads to SG death (*Fairchild et al., 2015*; *Lim and Fuller, 2012*). Though we noted significant SG death at the 16-SG stage in the course of our previous work, it was independent of nutrient conditions and thus was not the focus of the study (*Yang and Yamashita, 2015*).

In search of the cause of this 16-SG death, we discovered that it can be induced by ionizing radiation. When adult flies were exposed to ionizing radiation that causes DNA double strand breaks (DSBs), a dramatic induction of SG death was observed (*Figure 1B,C*). Dying SGs induced by ionizing radiation were detected by Lysotracker staining as described by previous studies, showing characteristic acidification of the entire cell (*Chiang et al., 2017*; *Yacobi-Sharon et al., 2013*; *Yang and Yamashita, 2015*). SG death in control and irradiated flies proceeded in the same manner, where all of the SGs within a cyst die simultaneously by becoming Lysotracker-positive (*Figure 1B*). Importantly, in contrast to starvation-induced SG death which was dependent on somatic cyst cell apoptosis, radiation-induced SG death was not suppressed by inhibiting cyst cell apoptosis (*Figure 1—figure supplement 1*), suggesting that radiation-induced SG death is a germ cell-intrinsic response.

The frequency of dying SG cysts peaked around 3 to 6 hr after irradiation and decreased by 24 hr post-irradiation (*Figure 1C*). Interestingly, we found that ionizing radiation robustly induces cell death at the 16-SG stage, although death of other stages (2-, 4-, 8 cell SGs) was also induced (*Figure 1C* and see below). This pattern of SG death held true regardless of the ionizing radiation dose (*Figure 1—figure supplement 2*). While testing multiple doses of ionizing radiation, we noticed that exposure to even a very low dose of ionizing radiation could dramatically induce death of 16-SGs. By measuring dose-dependent death of 16-SGs at six hours post-irradiation, we found

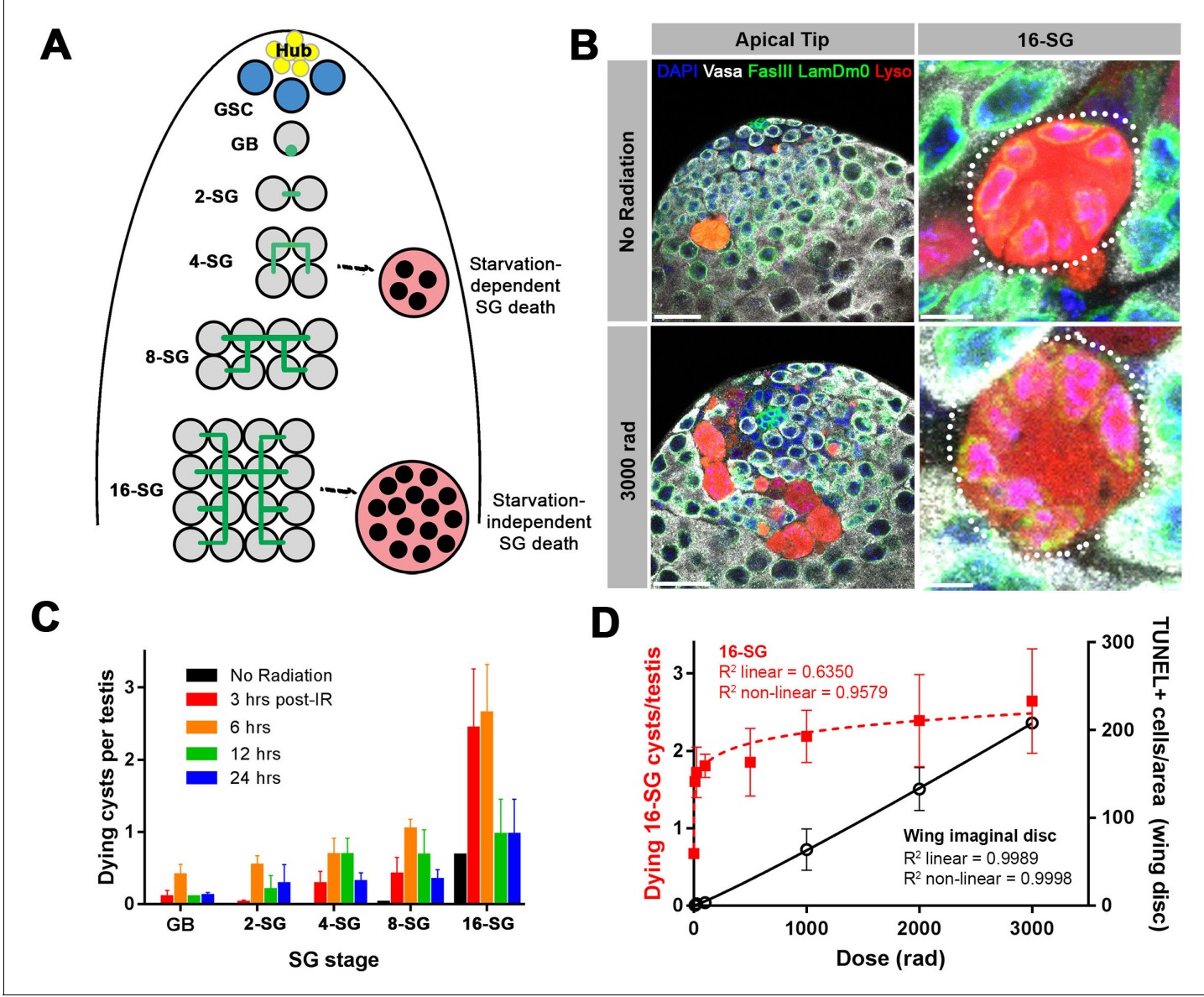

**Figure 1.** A high level of SG death in response to ionizing radiation. (**A**) Illustration of SG development and germ cell death in the *Drosophila* testis. (**B**) An example of the testis apical tip (left panels) with dying SGs marked by Lysotracker staining in control and irradiated flies. High magnification images of dying 16-SGs (dotted outline) are shown in right panels. Lysotracker (red), Vasa (white), FasIII and Lamin Dm0 (green), and DAPI (blue). Bars: 25 μm (left panels), 5 μm (right panels). (**C**) Quantification of dying SG cysts by stage from 3 to 24 hr after 3000 rad (Mean ±SD). n ≥ 17 testes, repeated in triplicate. It should be noted that the SG death frequency was scored as 'number of dying SG cysts at each stage per testis'. We have shown that the number of SG cysts is consistently ~5–6 cysts per stage per testis (*Yang and Yamashita, 2015*), justifying the use of 'number of dying SG cysts/testis' as a proxy for frequency of SG cyst death. (**D**) Number of Lysotracker-positive 16-SG cysts (red) and TUNEL-positive wing imaginal disc cells (black) 6 hr post-irradiation as a function of radiation dose (Mean ±SD). n ≥ 17 testes, and n ≥ 3 wing discs, repeated in triplicate. Best fit lines shown determined by non-linear regression.

DOI: https://doi.org/10.7554/eLife.27960.002

The following figure supplements are available for figure 1:

**Figure supplement 1.** Radiation-induced SG cyst death is independent of somatic cyst cell apoptosis.
DOI: https://doi.org/10.7554/eLife.27960.003
**Figure supplement 2.** SG death in response to ionizing radiation.
DOI: https://doi.org/10.7554/eLife.27960.004
**Figure supplement 3.** TUNEL staining to detect somatic cell death in response to ionizing radiation.
DOI: https://doi.org/10.7554/eLife.27960.005

that the 16-SG death induced by increasing radiation was a distinctly non-linear response, quickly reaching a plateau of ~3 dying 16-SG cysts per testis (*Figure 1D*). In comparison, cell death in the wing imaginal disc (*Figure 1—figure supplement 3*) followed a linear dose-response relationship where an increase in radiation resulted in a proportional increase in cell death (*Figure 1D*). These results demonstrate a remarkable sensitivity of 16-SGs to ionizing radiation compared to somatic cells.

## All SGs within a cyst die even when only a subset of cells exhibit detectable DNA damage

To gain insight into the cause of the unusual sensitivity of 16-SGs to DNA damage, we evaluated the response of SGs to ionizing radiation at a cell biological level. DSBs result in phosphorylation of the histone H2A variant (γ-H2Av), the *Drosophila* equivalent of mammalian γ-H2AX, reflecting the very early cellular response to DSBs (*Madigan et al., 2002*). Using an anti-γ-H2Av antibody (pS137), we confirmed that γ-H2Av can be robustly detected in SGs following a high dose of ionizing radiation (*Figure 2—figure supplement 1*). When a low dose of ionizing radiation was used (≤100 rad), we frequently observed 16-SG cysts in which only a subset of cells within the cyst exhibited detectable γ-H2Av signal but all 16 cells were Lysotracker-positive and dying (*Figure 2A*).

The fraction of γ-H2Av-positive SGs within each cyst increased gradually with increasing radiation dose irrespective of SG stage (*Figure 2B* and *Figure 2—figure supplement 2*), consistent with the linear nature in which ionizing radiation damages DNA molecules (*Ulsh, 2010*). However, Lysotracker staining showed that SGs within a cyst were always either all Lysotracker-positive or -negative (*Figure 2C*). These results suggest that while DNA damage is induced in individual SGs within a cyst proportional to the dose of radiation, cell death is induced in all of the SGs within the entire cyst, leading to elevated SG death that follows a non-linear response with increasing dose.

## The fusome is required for synchronized all-or-none SG death within the cysts

The above results led us to hypothesize that all SGs within a cyst might be triggered to die together even when only a subset of SGs within the cyst have detectable DNA damage, explaining the extremely high sensitivity of the germline to DNA damage. In *Drosophila* and other insects, the fusome is a germline-specific membranous organelle that connects the cytoplasm of germ cells within a cyst and mediates intracyst signaling amongst germ cells (*Lilly et al., 2000*; *Lin et al., 1994*). We speculated that if germ cell cysts undergo synchronized cell death by sharing the decision to die, the fusome might mediate the 'all-or-none' mode of SG death upon DNA damage.

To examine the role of the fusome in all-or-none SG death upon irradiation, we used RNAi-mediated knockdown of α-spectrin and a mutant of *hts*, core components of the fusome (*de Cuevas et al., 1996*; *Hime et al., 1996*; *Lilly et al., 2000*; *Lin et al., 1994*; *Yue and Spradling, 1992*) (*Figure 3—figure supplement 1*). Mutant and control flies were irradiated and their testes were stained with Lysotracker to identify dying SGs in combination with the lipophilic dye FM4–64 to mark cyst cell membranes, demarcating the boundaries of SG cysts (see methods) (*Chiang et al., 2017*). In control testes, 16-SG cysts were almost always found to be either completely Lysotracker-positive or -negative under all conditions tested as described above (*Figure 3A,B,E,F*), confirming our observation using fixed samples (*Figure 2*). Of particular importance, even at a lower dose of radiation (e.g. 100 rad) where only a subset of germ cells exhibit visible γ-H2Av staining (*Figure 2B*), the 16-SG cyst was either entirely Lysotracker-negative or -positive. In contrast, α-spectrin RNAi and *hts* mutant testes frequently contained 16-SG cysts with a mixture of individual Lysotracker-positive and -negative SGs (*Figure 3C,D,G,H*), suggesting that the all-or-none mode of SG death was compromised. Importantly, the mean fraction of 16-SG that died in response to radiation exposure at any dose was reduced when the fusome was disrupted (*Figure 3B,D,F,H*, black bars). These data show that the fusome is required for the coordinated death of all SGs within a cyst, and suggests that the intracyst communication increases overall SG death in response to radiation-induced DNA damage. Moreover, the fact that loss of germ cell communication in fusome mutants allows for the survival of some SGs strongly argues against the possibility that SGs without cytologically detectable γ-H2Av are sufficiently damaged to trigger cell death on their own and that SGs are individually dying. Instead, the death of SGs without detectable γ-H2Av in wild type/control can likely be attributed to a shared

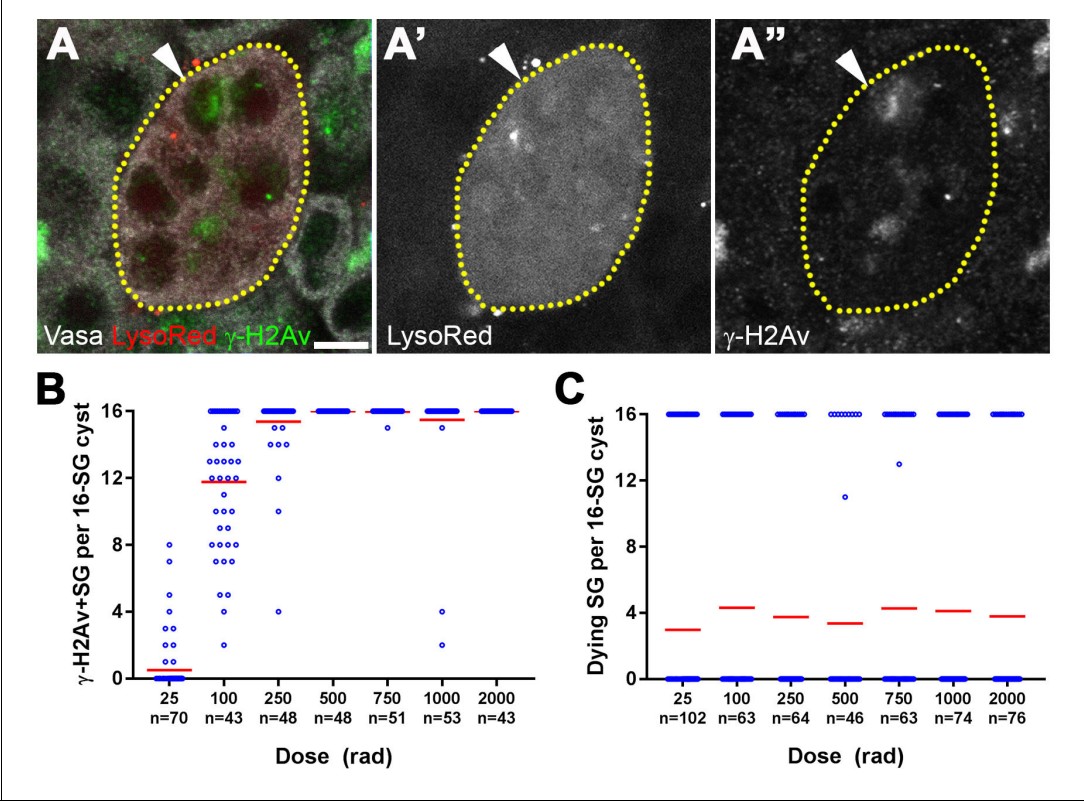

**Figure 2.** All SGs within a cyst die even when only a fraction of cells exhibit detectable DNA damage. (**A**) An example of a dying 16-SG cyst (yellow dotted outline) with only a subset of SGs containing detectable DNA damage (arrowhead). γ-H2Av (green), Lysotracker (red), Vasa (white). Bar: 5 μm. (**B**) Number of γ-H2Av-positive cells within each 16-SG cyst at various radiation doses. Blue circles, individual data points. Red line, mean. n = number of 16-SG cysts scored. (**C**) Number of Lysotracker-positive cells within each 16-SG cyst. Blue circles, individual data points. Red line, mean. n = number of 16-SG cysts scored.

DOI: https://doi.org/10.7554/eLife.27960.006

The following figure supplements are available for figure 2:

**Figure supplement 1.** γ-H2Av can be strongly detected in germ cells following irradiation.

DOI: https://doi.org/10.7554/eLife.27960.007

**Figure supplement 2.** All SG stages show gradual accumulation of γ-H2Av-positive cells with increasing radiation.

DOI: https://doi.org/10.7554/eLife.27960.008

death signal from other cells, as blockade of intercellular communication in fusome mutants allows for their survival.

It should be noted that the fusome mutants appeared to maintain ring canals, as evidenced by the intact ring shape of Pavarotti-GFP, a marker for ring canals (*Minestrini et al., 2002*) (*Figure 3—figure supplement 2*). Therefore, intracyst communication triggering SG death is likely mediated by the fusome, rather than physical openings between SGs within the cyst (see Discussion). Consistent with this, somatic follicle cells of the egg chamber, which are known to be connected by ring canals without a fusome (*McLean and Cooley, 2013a*), were observed to die individually by becoming positive for cleaved caspase (Dcp-1) (*Figure 3—figure supplement 3A*). Additionally, these cells did not exhibit extreme sensitivity to low doses of radiation and displayed an essentially linear dose-response (*Figure 3—figure supplement 3B,C*).

## The mitochondrial proteins HtrA2/Omi and Endonuclease G are required for all-or-none SG death

The above results suggest that intercellular communication mediated by the fusome plays a critical role in allowing for all-or-none commitment of SGs to death or survival. Based on these results, we

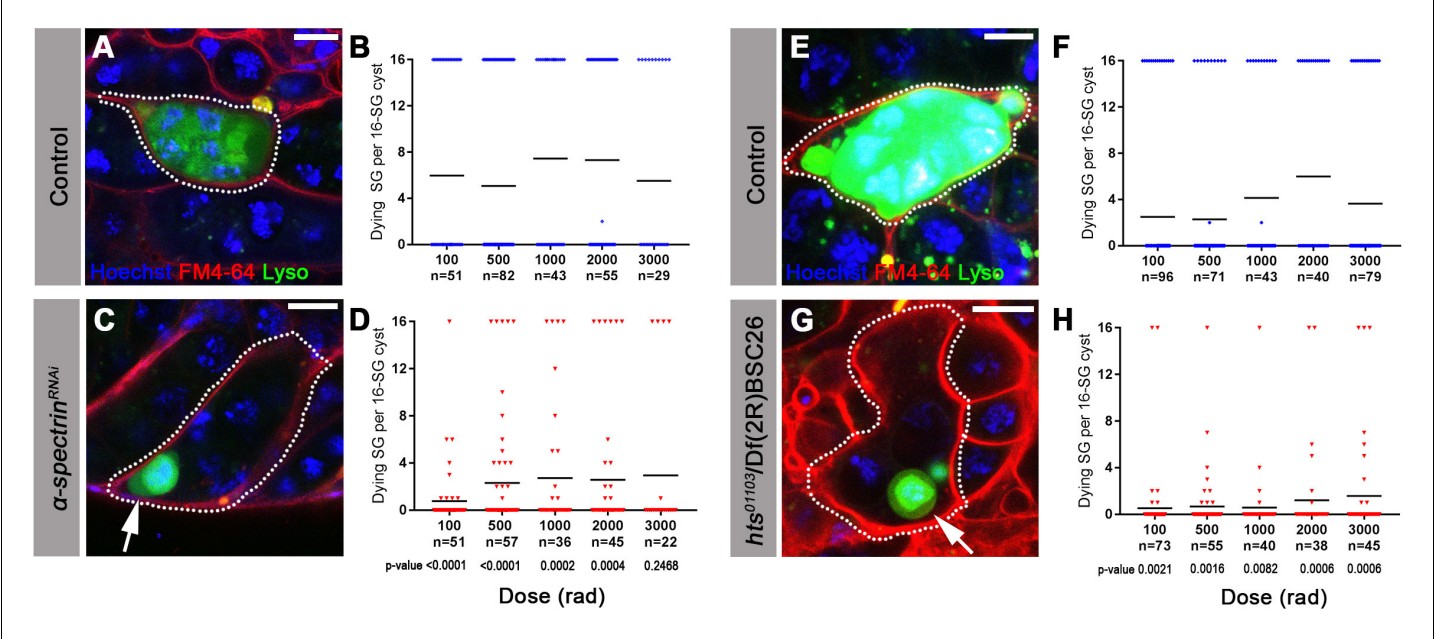

**Figure 3.** The fusome is required for synchronized all-or-none SG death within a cyst. (**A**) A Lysotracker-positive 16-SG cyst (green, dotted outline) in unfixed control testes, with cyst borders marked by FM4–64 (red) and SG nuclei marked by Hoechst 33342 (blue). Bar: 7.5 µm. (**B**) Number of Lysotracker-positive cells within each 16-SG cyst of control testes at varying radiation doses. Black line, mean. n = number of 16-SG cysts scored. (**C**) A 16-SG cyst (dotted outline) in *nos-gal4* >UAS-*α-spectrin*$^{RNAi}$ testes containing a single Lysotracker-positive SG (arrowhead). Bar: 10 µm. (**D**) Number of Lysotracker-positive cells within each 16-SG cyst of *nos-gal4* >UAS-*α-spectrin*$^{RNAi}$ testes at varying radiation doses. Black line, mean. n = number of 16-SG cysts scored. P-values (comparing the corresponding radiation doses between control and mutant) determined by chi-squared test (See methods). (**E, F**) *hts*$^{01103}$/+ control testes. Bar: 5 µm. (**G, H**) *hts*$^{01103}$/Df(2R)BSC26 mutant testes. Bar: 7.5 µm.

DOI: https://doi.org/10.7554/eLife.27960.009

The following source data and figure supplements are available for figure 3:

**Source data 1.** Number of dying SGs per 16-SG cyst in control, *α-spectrin*$^{RNAi}$ and *hts* mutant testes in response to irradiation.
DOI: https://doi.org/10.7554/eLife.27960.013
**Figure supplement 1.** Validation of fusome elimination in *hts* mutant and *α-spectrin*$^{RNAi}$ testes.
DOI: https://doi.org/10.7554/eLife.27960.010
**Figure supplement 2.** SG ring canals are maintained in fusome mutants.
DOI: https://doi.org/10.7554/eLife.27960.011
**Figure supplement 3.** Follicle cell death in response to irradiation.
DOI: https://doi.org/10.7554/eLife.27960.012

hypothesized that a signal to promote cell death exists that is rapidly transmitted from damaged SGs to others via their intercellular connections.

It has previously been shown that germ cell death in the *Drosophila* testis is non-apoptotic, and depends on mitochondria-associated factors (*Yacobi-Sharon et al., 2013*). The *Drosophila* homolog of the mitochondrial serine protease HtrA2/Omi is cleaved and released from the mitochondrial compartment as part of the mitochondria-associated death pathway to promote cell death in response to nuclear DNA damage (*Igaki et al., 2007*; *Khan et al., 2008*; *Tain et al., 2009*; *Vande Walle et al., 2008*). The catalytic function of released HtrA2/Omi is known to control the death of SGs in the testis. Yacobi-Sharon et al. further showed that Endonuclease G (EndoG) is also involved in germ cell death (*Yacobi-Sharon et al., 2013*). EndoG normally resides in mitochondria but is released to promote degradation of nuclear chromatin and induce cell death (*Widlak and Garrard, 2005*).

In *HtrA2/Omi* mutant flies, we frequently observed a mix of Lysotracker-positive and -negative SGs within a single cyst, similar to what was seen in fusome mutants (*Figure 4A*). Disruption of the all-or-none mode of SG death within the cyst was observed at any dose of radiation tested (*Figure 4B*). Disrupted all-or-none SG death occurred in both heterozygous (*Omi*$^{Δ1}$/+) and

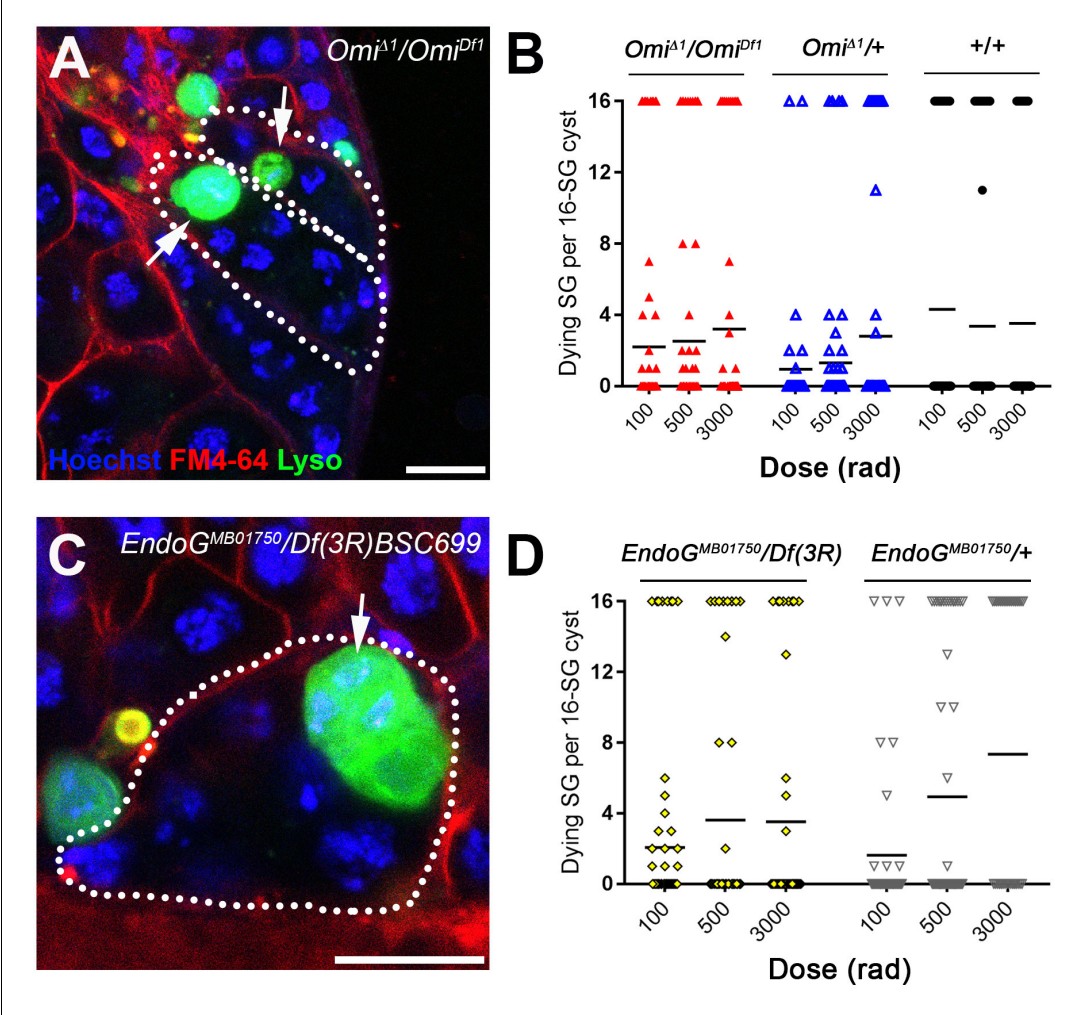

**Figure 4.** The mitochondrial proteins HtrA2/Omi and Endonuclease G are required for all-or-none SG death. (**A**) SG cysts (dotted outlines) in $Omi^{\Delta 1}$/$Omi^{Df1}$ mutant testes containing individual Lysotracker-positive SGs (arrows). Hoechst 33342 (blue), FM 4–64 (red), Lysotracker (green). Bar: 10 μm. (**B**) Number of Lysotracker-positive SGs in each 16-SG cyst in $Omi^{\Delta 1}$/$Omi^{Df1}$, $Omi^{\Delta 1}$/+, and wild type testes. Black line, mean. n ≥ 43 cysts per dose. (**C**) 16-SG cyst (dotted outline) in $EndoG^{MB07150}$/Df(3R)BSC699 testes containing individual Lysotracker-positive SGs. Hoechst 33342 (blue), FM 4–64 (red), Lysotracker (green). Bar: 10 μm. (**D**) Number of Lysotracker-positive SGs in each 16-SG cyst in $EndoG^{MB07150}$/Df(3R)BSC699 and $EndoG^{MB07150}$/+ testes. Black line, mean. n ≥ 37 cysts per dose.

DOI: https://doi.org/10.7554/eLife.27960.014

The following source data is available for figure 4:

**Source data 1.** Number of dying SGs per 16-SG cyst in control, *Omi* and *Endo G* mutant testes in response to irradiation.

DOI: https://doi.org/10.7554/eLife.27960.015

transheterozygous ($Omi^{\Delta 1}$/$Omi^{Df1}$) conditions, consistent with the previous report that heterozygous conditions exhibit haploinsufficiency in inducing SG death (*Yacobi-Sharon et al., 2013*). In contrast, wild type control 16-SG cysts maintained their all-or-none mode of SG death (*Figure 4B*). Likewise, a loss-of-function $EndoG^{MB07150}$ mutant allele (*DeLuca and O'Farrell, 2012*) also showed disruption of all-or-none SG death in both heterozygous ($EndoG^{MB07150}$/+) and transheterozygous ($EndoG^{MB07150}$/Df(3R)BSC699) conditions (*Figure 4C,D*). Taken together, these data show the involvement of mitochondrial cell death pathway components HtrA2/Omi and EndoG in the all-or-none mode of SG death. Considering that these proteins are released from mitochondria to induce cell death, it is tempting to speculate that these proteins (or their downstream molecules/signals) are shared among SGs via the fusome to trigger synchronized SG death.

### *p53* and *mnk/chk2* do not regulate the all-or-none mode of SG death

The DNA damage response is a highly conserved pathway controlling cell death and DNA repair (*Ciccia and Elledge, 2010*; *Song, 2005*). We thus examined the potential involvement in radiation-induced SG death of the universal DNA damage response pathway components, *mnk/chk2* and *p53*, whose conserved function in DNA damage response in *Drosophila* has been shown (*Brodsky et al., 2004*; *Peters et al., 2002*). By using well-characterized loss-of-function alleles (*mnk*[6006] and *p53*[5A-1-4]) (*Takada et al., 2003*; *Wichmann et al., 2006*; *Xie and Golic, 2004*), we found that these mutants broadly suppress SG death at high and low doses of radiation (*Figure 5D*). However, SG death in these mutants maintained an all-or-none pattern (*Figure 5A–C,E*). These results suggest that while *p53* and *mnk/chk2* may contribute to SG death via their general role in controlling the DNA damage response as has been described in somatic cells, they do not play a role in mediating the all-or-none pattern of SG death within a cyst that is unique to interconnected germ cells.

Consistent with the idea that neither *mnk/chk2* or *p53* is responsible for a germline-specific, all-or-none mode of cell death in response to DNA damage, Mnk/Chk2 or p53 was barely upregulated in the germline in response to a low dose of radiation (100 rad), which is sufficient to induce robust SG death. Using a polyclonal anti-Mnk/Chk2 antibody (*Takada et al., 2015*), we barely detected any signal in germ cells, although some level of signal was seen in the surrounding somatic cyst cells (*Figure 5—figure supplement 1*). Likewise, a p53 transcriptional reporter (*Wylie et al., 2014*) showed only a slight increase in the signal in response to a low dose of radiation (*Figure 5—figure supplement 2*). Even at a high dose, robust expression of the reporter was observed only at 24 hr after irradiation, much later than the peak of SG death, which typically happens within a few hours. Collectively, these data (i.e. *mnk/chk2* and *p53* mutants maintaining all-or-none cell death and the lack of robust Mnk/Chk2 and p53 expression in response to DNA damage) indicate that expression of p53 or Mnk/Chk2 in SGs is unlikely to account for the all-or-none mode of germ cell death in response to DNA damage.

### Increasing connectivity of SGs increases sensitivity to DNA damage

The above results suggest that the ability of SGs to trigger cell death in response to DNA damage is facilitated by the sharing of death-promoting signals amongst SGs within a cyst, killing SGs that are not sufficiently damaged to commit to cell death on their own. This sharing of death-promoting signals is mediated by and dependent on the fusome, which facilitates intracyst communication. If this is the case, it would be predicted that increasing the connectivity of a SG cyst (the number of interconnected SGs within the cyst) would increase its sensitivity to DNA damage, because increased SG number per cyst will increase the probability of any cells being sufficiently damaged to trigger death. Indeed, as mentioned above, we observed a trend of 16-SG cysts dying more frequently than 2-, 4-, or 8-SGs (*Figure 1C*, *Figure 1—figure supplement 2*). By plotting cell death frequency of all SG stages as a function of increasing radiation dose (*Figure 6A*), it becomes clear that the sensitivity of SG cysts correlates with their connectivity, where 8-SGs are less sensitive than 16-SGs but more sensitive than 4-SGs and so on. Interestingly, single-celled GBs, the immediate daughter of GSCs that have not formed any intercellular connections, exhibited an essentially linear increase in death in response to radiation dose, which is reminiscent of somatic imaginal disc cells or follicle cells (*Figure 1D*, *Figure 3—figure supplement 3*). These results support the idea that germ cell connectivity plays a key role in increasing the sensitivity of the germline to DNA damage.

## Discussion

Our present study may provide a link between two long-standing observations in germ cell biology: (1) the broad conservation of intercellular connectivity (cyst formation) of germ cells and (2) the sensitivity of the germline to DNA damage. The purpose for germ cell cyst formation outside the meroistic ovary (i.e. nursing mechanism) remained unclear. Our study in the *Drosophila* male germline shows that the connectivity of germ cells can serve as a key mechanism for their ability to robustly induce cell death, providing an explanation for cyst formation outside of the oocyte nursing mechanism. However, there are many cell types that are known to be connected to sibling cells, where synchronized cell death among the connected cells is not observed or does not make any biological sense. For example, early-stage embryos of *Drosophila* develop as syncytia, where all nuclei are

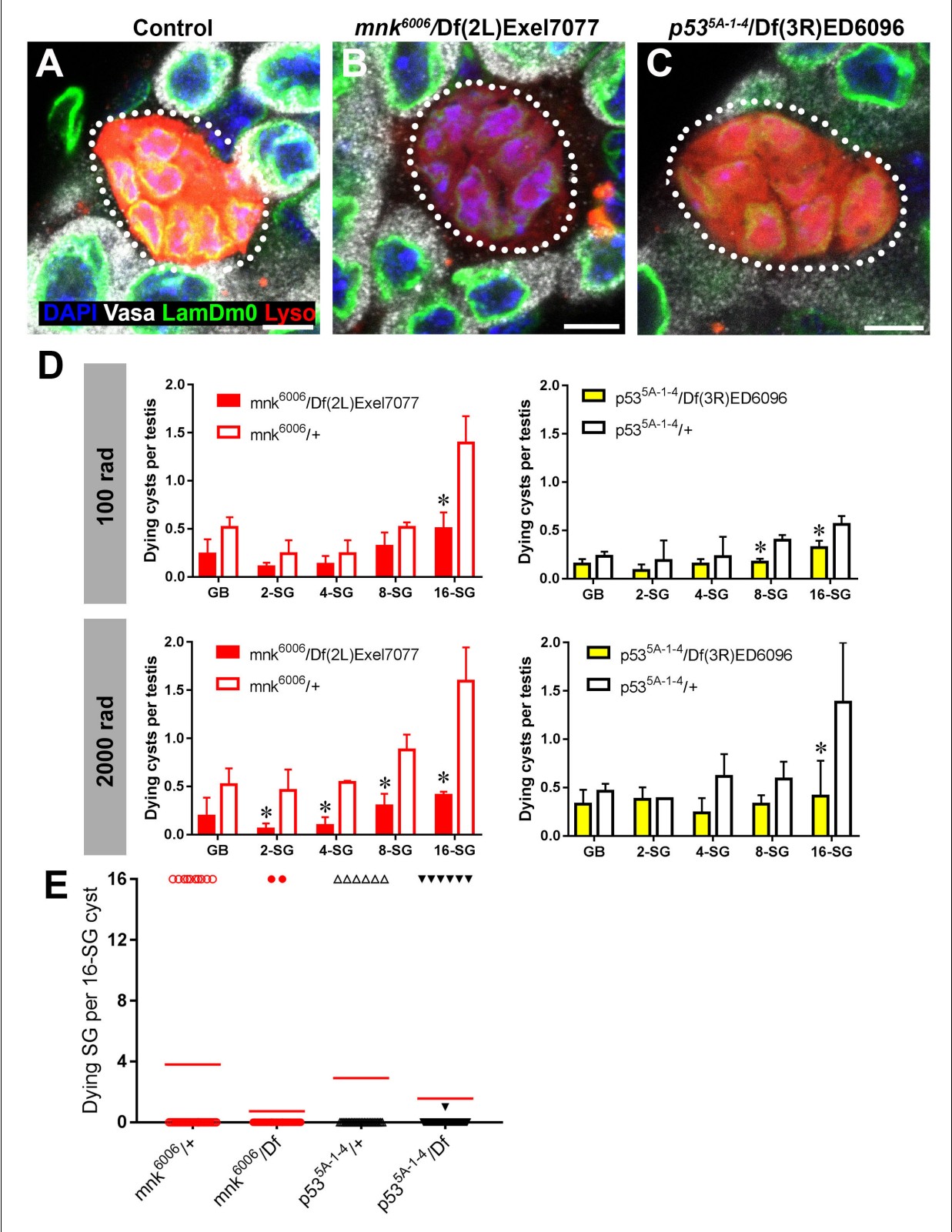

**Figure 5.** *p53* and *mnk/chk2* suppress SG death but do not regulate the all-or-none mode of SG death. (**A–C**) Examples of dying 16-SGs (dotted outline) in wild-type (**A**), *mnk⁶⁰⁰⁶*/Df(2L)Exel7077 mutant (**B**), and *p53⁵ᴬ⁻¹⁻⁴*/Df(3R)ED6096 mutant (**C**) testes. Lysotracker (red), Lamin Dm0 (green), DAPI (blue) and Vasa (white). Bars: 5 µm. (**D**) SG cyst death by stage in *mnk/chk2* and *p53* mutants 6 hr after irradiation with 100 rad and 2000 rad (Mean ±SD, p-value *<0.05 t-test). Fixed, stained samples were used for scoring. Testes sample n ≥ 11 for each genotype, repeated in triplicate. (**E**) Number of

*Figure 5 continued on next page*

*Figure 5 continued*

Lysotracker-positive SG per 16-SG cyst in *mnk/chk2* and *p53* mutants following 100 rad. Red line, mean. Unfixed samples stained with Lysotracker, FM 4–64, and Hoechst 33342 were used for scoring.

DOI: https://doi.org/10.7554/eLife.27960.016

The following source data and figure supplements are available for figure 5:

**Source data 1.** Number of dying SGs per 16-SG cyst in control, *mnk/chk2* and *p53* mutant testes in response to irradiation.

DOI: https://doi.org/10.7554/eLife.27960.019

**Figure supplement 1.** Expression of Mnk/Chk2 in response to ionizing radiation.

DOI: https://doi.org/10.7554/eLife.27960.017

**Figure supplement 2.** Expression of p53 reporter in response to ionizing radiation.

DOI: https://doi.org/10.7554/eLife.27960.018

within a connected cytoplasm. The *C. elegans* germline is topologically similar to *Drosophila* early embryos in that all cells share the same cytoplasm. In these two examples, damaged nuclei die individually and synchronized cell death is not observed (*Lettre et al., 2004*; *Takada et al., 2003*). In *Drosophila* oogenesis, 16 interconnected cystocytes have distinct fates (one oocyte and 15 nurse cells), and it would not be beneficial to kill the oocyte when just one dispensable nurse cell is damaged. Moreover, nurse cells undergo programmed apoptosis later in oogenesis to finalize the cytoplasmic transport into the oocyte, yet this apoptosis does not kill oocytes. These examples have two important implications. First, openness via ring canals/intercellular bridges alone is likely insufficient to mediate synchronized cell death. Second, there are likely additional purposes for intercellular connectivity to be discovered other than the sharing of death-promoting signals.

Regarding the first point that physical openness alone is likely insufficient to induce cell death in undamaged sibling cells, we speculate that the fusome plays a key role in facilitating intercellular communication of death-promoting signals. Because fusome mutants apparently maintain cytoplasmic openings (ring canals) between SGs (*Figure 3—figure supplement 2*), it suggests that the intercellular bridges/openings are not sufficient to allow intracyst communication of death-

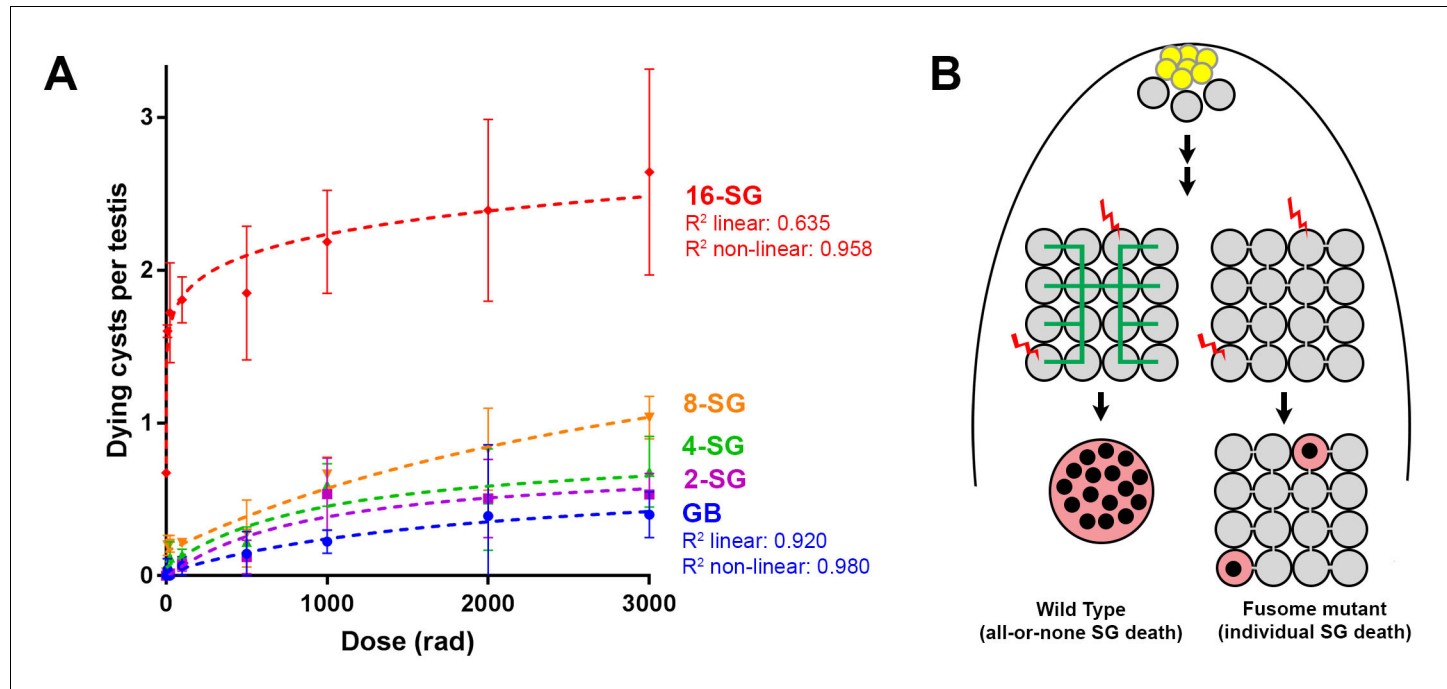

**Figure 6.** Increasing connectivity confers higher sensitivity to DNA damage. Dose-dependent SG death in 2-, 4-, 8-, and 16-SG cysts (Mean ±SD). Best fit lines shown determined by non-linear regression. n ≥ 17 testes, repeated in triplicate. (A) (B) Model of SG death enhanced by connectivity.

DOI: https://doi.org/10.7554/eLife.27960.020

promoting signals. Consistent with this idea, we found that follicle cells of the egg chamber do not exhibit synchronized cell death in response to irradiation (*Figure 3—figure supplement 3A*), even though these cells are known to maintain intercellular bridges/ring canals among multiple sibling cells, through which molecules (e.g. GFP) can be transported (*McLean and Cooley, 2013a*). The lack of synchronized cell death in the follicle cells coincided with a proportional/linear increase in cell death with the increasing dose of radiation (*Figure 3—figure supplement 3B,C*). Therefore, we propose that the fusome functions to transmit the death-triggering signals, similar to its known role in cell cycle synchronization within the germ cell cyst (*Lilly et al., 2000*)(*Figure 6B*).

We imagine two possibilities to explain how the signals are shared within a cyst, leading to cell death: when one SG within a cyst decides to die, this 'decision of death' might be sent to all the other SGs within the cyst, leading to all-or-none SG death. Alternatively, the signal shared among SGs may be 'additive' in nature. In such a scenario, even when none of the individual SGs have sufficient DNA damage to trigger cell death on its own, addition of all damage signaling within the cyst might reach a level sufficient to induce cell death. Either way, such responses would effectively lower the threshold of DNA damage per cell needed to trigger germ cell death. Consequently, the sharing of signals between SGs through intercellular connections increases their likelihood to die following DNA damage, explaining the unusually high sensitivity of the germline to DNA damage.

According to this model, higher connectivity would confer higher sensitivity to DNA damage: as the connectivity increases, more cells would contribute to detecting any DNA damage the germline may be experiencing. Indeed, our data show a positive correlation between sensitivity to radiation and the increasing connectivity of SG cysts (*Figure 6A*). Remarkably, the fact that single-celled, unconnected GBs exhibit an essentially linear death response to increasing radiation suggests that individual germ cells do not necessarily have an intrinsically different DNA damage response that accounts for their high sensitivity to DNA damage (*Figure 6A*).

A connectivity-based increase in sensitivity to DNA damage also has an important implication in the development of multicellular organisms. To pass on genomes to the next generation, it is critically important for germ cells to have the most stringent mechanisms to prevent deleterious mutations. However, as genome size increases in multicellular organisms, ubiquitously increasing the stringency of genome quality control could result in a high rate of cell death in all tissues, which could compromise the development or survival of organisms. Thus, a multicellular organism may require differential sensitivities to DNA damage between the soma and the germline: a more sensitive genome surveillance mechanism to protect the germline, whereas the priority of the soma shifting toward survival to support development/maintenance of somatic organs. A connectivity-based increase in sensitivity to DNA damage might be a simple method for multicellular organisms to achieve drastically different sensitivities to DNA damage between the soma and germline without having to alter intrinsic damage response pathways, although additional germline-specific DNA damage response mechanisms cannot be excluded. We speculate that one reason germ cell connectivity has arisen during evolution and been so widely conserved might be to confer higher sensitivity to DNA damage specifically in the germline. It awaits future studies to understand whether germ cells from other organisms exhibit high sensitivity to DNA damage in a manner dependent on intra-cyst communication.

## Materials and methods

### Fly husbandry and strains

All fly stocks were raised on standard Bloomington medium at 25°C, and young flies (0 to 2 day old adults) were used for all experiments unless otherwise noted. The following fly stocks were used: *hts*[01103] (*Yuan et al., 2012*), *nos-gal4* (*Van Doren et al., 1998*), UAS-$\alpha$-*spectrin*[RNAi] (TRiP. HMC04371), *c587-gal4* (*Decotto and Spradling, 2005*), UAS-*Diap1*, *EndoG*[MB07150] (*DeLuca and O'Farrell, 2012*) (obtained from the Bloomington Stock Center), *p53*[5A-1-4] (*Wichmann et al., 2006*; *Xie and Golic, 2004*), Df(2R)BSC26, Df(3R)ED6096, Df(2L)Exel7077, Df(3R)BSC699 (obtained from the Bloomington Stock Center), *p53RE-GFP-nls* reporter (*Wylie et al., 2014*) (a gift of John Abrams, University of Texas Southwestern Medical Center), *Omi*[Δ1], *Omi*[Df1] (*Tain et al., 2009*; *Yacobi-Sharon et al., 2013*) (a gift of Eli Arama, Weizmann Institute of Science), *mnk*[6006] (*Takada et al.,*

*2003*) (a gift of William Theurkauf, University of Massachusetts Medical School), Ubi-Pavarotti-GFP (*Minestrini et al., 2002*) (a gift of David Glover, University of Cambridge).

## Immunofluorescence staining and microscopy

Immunofluorescence staining of testes was performed as described previously (*Cheng et al., 2008*). Briefly, testes were dissected in PBS, transferred to 4% formaldehyde in PBS and fixed for 30 min. Testes were then washed in PBS-T (PBS containing 0.1% Triton-X) for at least 60 min, followed by incubation with primary antibody in 3% bovine serum albumin (BSA) in PBS-T at 4°C overnight. Samples were washed for 60 min (three 20 min washes) in PBS-T, incubated with secondary antibody in 3% BSA in PBS-T at 4°C overnight, washed as above, and mounted in VECTASHIELD with DAPI (Vector Labs). The following primary antibodies were used: mouse anti-Adducin-like 1B1 (*hu-li tai shao –* Fly Base) [1:20; Developmental Studies Hybridoma Bank (DSHB); developed by H.D. Lipshitz]; mouse anti-alpha-spectrin 3A9 (1:20; DSHB; developed by R. Dubreuil, T. Byers); rat anti-vasa (1:50; DSHB; developed by A. Spradling), rabbit anti-vasa (1:200; d-26; Santa Cruz Biotechnology), mouse anti-Fasciclin III (1:200; DSHB; developed by C. Goodman), anti-LaminDm0 (1:200; DSHB; developed by P. A. Fisher), rabbit anti-γ-H2AvD pS137 (1:100;Rockland), rabbit anti-Mnk (1:100; courtesy of Saeko Takada), rabbit anti-Cleaved Drosophila Dcp-1 (Asp216) (1:200; Cell Signaling Technology). Images were taken using a Leica TCS SP8 confocal microscope with 63x oil-immersion objectives (NA = 1.4) and processed using Adobe Photoshop software. For detection of germ cell death, testes were stained with Lysotracker Red DND-99 in PBS (1:1000) for 30 min prior to formaldehyde fixation. Stages (GB, 2-, 4-, 8-, and 16-SGs) of dying SGs were identified by counting the number of nuclei within the cyst visualized by Lamin Dm0 and DAPI (*Chiang et al., 2017*). Note that the number of 'stageable' dying SGs underrepresents the total population of dying SGs, because nuclear structures disintegrate during later phases of cell death, making it impossible to count the number of SGs within a dying cyst (*Chiang et al., 2017*). Such 'unstageable' SGs were not included in the scoring in this study.

For observation of unfixed samples, testes were dissected directly into PBS and incubated in the dark with the desired dyes for 5 min, mounted on slides with PBS and imaged within 10 min of dissection. The dyes used in live imaging are: Lysotracker Red DND-99 (1:200) or Lysotracker Green DND-26 (1:200) (Thermo Fisher Scientific), Hoechst 33342 (1:200), and FM4-64FX in PBS (1:200) (Thermo Fisher Scientific). SG stage (2-, 4-, 8-, or 16-SG) was assessed by number of Hoechst-stained nuclei. Note that the scoring of dying SGs is not directly comparable between fixed and unfixed samples (for example, results shown in *Figure 1* vs. *Figure 2*), due to the difference in the method of SG staging and timing.

## Ionizing radiation

For radiation doses of 25–250 rad, a $^{137}$Cs source was used with a dose rate of approximately 100 rad per minute. Additionally, for radiation doses 100 rad and above, a Philips RT250 model or Kimtron Medical IC-320 orthovoltage unit was used (dose rates of 200 and 400 rad per minute respectively). Dosimetry was carried out using an ionization chamber connected to an electrometer system directly traceable to National Institute of Standards and Technology calibration. The relative biological effectiveness of $^{137}$Cs and x-ray sources is comparable at lower doses (*Fu et al., 1979*), and experiments were repeated with both sources for 100–250 rad, which yielded essentially the same results irrespective of radiation source.

## TUNEL assay

Larval heads with imaginal discs attached were dissected from third instar larvae into PBS, then fixed in 4% formaldehyde in PBS for 30 min. Samples were then washed in PBS-T (PBS containing 0.1% Triton-X) for at least 20 min, transferred to 100% methanol for 6 min with rocking, and washed again for at least 20 min in PBS-T. TUNEL assay was then carried out according to manufacturer's instructions using a Millipore ApopTag Red In Situ Apoptosis Detection Kit (S7165). Following washes with PBS-T for 20 min, wing imaginal discs were dissected from larval heads and mounted in VECTASHIELD with DAPI (Vector Labs).

## Dose-response best fit regressions

Best fit functions and lines for radiation dose-cell death response curves were generated by using GraphPad Prism seven and the means-only values at all doses. Non-linear regressions were determined using a four-parameter logistic curve with no constraints on bottom, top, or hillslope and >1500 iterations. Standard linear regression was performed using cell death as a function of radiation dose. Goodness of fit, unadjusted $R^2$ value, was determined by 1.0 less the ratio of the regression sum of squares to the total sum of squares, $1 - SS_{reg}/SS_{tot}$.

## Statistics for comparison of 16-SG death distributions

Distribution of number of dying SGs per 16-SG cyst between mutant and control conditions at every dose of radiation were compared using a $2 \times 3$ Pearson's chi-squared test. Data were transformed into three categories: 0 SGs dying, all 16 SGs dying, or 1–15 (partial cyst) SGs dying.

## Acknowledgements

We thank Drs. Saeko Takada, William Theurkauf, Eli Arama, David Glover, and John Abrams for reagents. Bloomington Stock Center, the Developmental Studies Hybridoma Bank for reagents, the University of Michigan Experimental Irradiation Core for radiation experiments, the University of Michigan Center for Consulting for Statistics, Computing, and Analytics Research for help with statistical analyses, Drs. Lei Lei, Sue Hammoud and Yamashita lab members for comments on the manuscript, Dr. Lynn Cooley for sharing unpublished results. This work was supported by the University of Michigan Medical Scientist Training Program, the University of Michigan Career Training in Reproductive Biology training grant [T32 HD079342] and NIH Fellowship [F30 AG050398-01] to KL, and by the Howard Hughes Medical Institute to YY.

## Additional information

### Competing interests

Yukiko M Yamashita: Reviewing editor, *eLife*. The other author declares that no competing interests exist.

### Funding

| Funder | Grant reference number | Author |
| --- | --- | --- |
| Howard Hughes Medical Institute | | Yukiko M Yamashita |
| National Institute of General Medical Sciences | R01 GM118308-01 | Yukiko M Yamashita |
| National Institute on Aging | F30 AG050398-01A1 | Kevin L Lu |

The funders had no role in study design, data collection and interpretation, or the decision to submit the work for publication.

### Author contributions

Kevin L Lu, Conceptualization, Formal analysis, Funding acquisition, Investigation, Writing—original draft, Writing—review and editing; Yukiko M Yamashita, Conceptualization, Supervision, Funding acquisition, Investigation, Writing—original draft, Project administration, Writing—review and editing

### Author ORCIDs

Yukiko M Yamashita (iD) https://orcid.org/0000-0001-5541-0216

### Decision letter and Author response

Decision letter https://doi.org/10.7554/eLife.27960.022
Author response https://doi.org/10.7554/eLife.27960.023

## Additional files

### Supplementary files

• Transparent reporting form
DOI: https://doi.org/10.7554/eLife.27960.021

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
