## [Decision Letter]

Thank you for submitting your article "Germ cell connectivity enhances cell death in response to DNA damage in *Drosophila* testis" for consideration by *eLife*. Your article has been reviewed by three peer reviewers, and the evaluation has been overseen by a Reviewing Editor, Hugo J Bellen, and K VijayRaghavan as the Senior Editor. The following individual involved in review of your submission has agreed to reveal his identity: Allan C Spradling (Reviewer #2).

The reviewers have discussed the reviews with one another and the Reviewing Editor has drafted this decision to help you prepare a revised submission. The Reviewing editor has assembled the following comments based on the reviewers' reports, which are considered essential concerns that you will need to address in a revised submission.

*Reviewer #1:*

In this manuscript, the authors examine a potential link between the extraordinary sensitivity of germ cells to DNA damage and their syncytial mode of development. Using irradiated developing *Drosophila* male germ cells as a model, they show that the presence of damaged spermatogonial cells within a syncytial cyst results in death of all cells within the cyst. They show that this coordinated "all-or-none" germ cell death relies on the presence of the adducin-like protein Hts and spectrin, which are components of the fusome (an ER-like organelle that interconnects developing spermatocytes via stable intercellular bridges), as well as the mitochondrial protease HtrA2/Omi. In addition, the authors argue, based on the presence of higher numbers of dying 16-cell spermatogonial cysts, that higher connectivity leads to greater sensitivity to DNA damage, suggesting a possible evolutionary advantage to syncytial germ cell development. Overall, the manuscript is well written and the experiments are nicely executed.

*Reviewer #2:*

This paper presents a fascinating, original idea regarding the origin of germ cell radiation sensitivity. The authors connect DNA damage sensitivity to the intercellular bridges that join sets of male or female germ cells into "germline cysts" for much of gamete development. Their hypothesis is strongly supported in the case of pre-meiotic 2,4,8 and especially 16-cell SG cysts. Disruption of cyst connectivity using fusome mutants or interference with the proposed damage signaling pathway using HtrA2/Omi mutants causes effects predicted by the hypothesis and the signaling mechanism. The existence of a novel group mechanism elevating damage sensitivity at least in pre-meiotic cysts is of strong interest in its own right. Shortly after these stages, during meiosis, extensive DNA damage occurs to stimulate recombination, raising the question of how the propagation of damage signals in avoided. If interconnected pre-meiotic male germ cells can transmit apoptotic signals between cells, how do interconnected female germ cells avoid damaging the oocyte during their own apoptotic process that accompanies "nurse cell dumping? "

The argument that increased connectivity leads to increased sensitivity to DNA damage is not convincing. The authors find that there are more dying 16-cell cysts than dying gonialblasts or cysts at earlier stages. Surprisingly, they do not take into account the frequency of dying cysts at different stages relative to the abundance of cysts at these stages within the testis. 16-cell spermatocyte cysts are known to persist for ~90 hours prior to undergoing meiotic division. Since the entire period prior to meiotic division is thought to encompass ~5 days (~120 hrs), simple math would dictate that there are likely ~3x more 16-cell cysts than cysts at earlier stages of sperm development. In fact, adult testes clearly contain a preponderance of 16-cell cysts, with smaller numbers of cysts at earlier stages. Thus, the observation that there are more dying cysts at the 16-cell stage could simply be due to the fact that there are more cysts at this stage. The authors' numbers would seem to agree with this. For example, there are approximately 3x more dying 16-cell cysts at 3 and 6 hrs post-irradiation (Figure 1) than there are dying cysts at earlier stages. I strongly suggest that the authors quantify the numbers of cysts at different stages in wild-type control testes (or find references where this has been quantified), as their conclusions in this regard clearly overstep the data presented (there is no denominator) and in my view do not make sense. Moreover, support for the idea that the degree of connectivity of germ cells and communication of a DNA damage signal are not needed for increased sensitivity of developing germ cells to DNA damage is based on what is known about *C. elegans* germline development: although *C. elegans* germ cells develop within a syncytium, germ cells that are damaged by irradiation cellularize and undergo apoptosis as individual cells; DNA damage does not induce massive cell death within the large syncytium that comprises a major proportion of the *C. elegans* gonad. This further argues that there is no evolutionary requirement for increased connectivity in promoting germ cell death in response to DNA damage. The novelty of what the authors show for the *Drosophila* male germline is that connectivity leads to death of all of the cells within a cyst and that this is apparently due to sharing of a death signal among the syncytial SG cells.

Is the fusome itself required for the coordinated death of all SGs within a cyst or is it that the interconnectivity between cysts is what is important and that fusome loss compromises interconnectivity? How does knockdown of α-spectrin or mutation of *hts* influence ring canals and the ability of small molecules and proteins to move between cells within a cyst? All this may be well established or known, however, as written it was not clear to us. Please clarify this in the text. Or conduct experiment that demonstrate diffusion or lack thereof within the normal and *hts* mutant cysts.

The biggest current weakness of the manuscript is the claim that apoptosis signaling within the cyst can explain the "the extreme sensitivity" of germ cells to DNA damage. First, it is not clear that germ cell damage hyper-sensitivity has been rigorously established as what data exist derive heavily from observations of humans treated with chemotherapeutic agents. But even if germ cells are hypersensitive as a group, the authors need to explain how a mechanism that acts at relatively rare stages could strongly affect the bulk sensitivity of germ cells which are not mostly at pre-meiotic cyst stages. Consequently, the authors should more thoroughly investigate how long during meiosis and the remaining time that clusters remain interconnected does sensitivity stay elevated and all or nothing apoptosis patterns persist. If the effects are actually widespread, then this apparent difficulty would be resolved.

Similarly, if a yet unknown mechanism leads to communication of 'hypothetical death signal' between cells in cysts one would expect that other, somatic syncytia, such as the early embryos or polynucleated muscles, should be similarly sensitive. These experiments may exist in the literature or otherwise should be included.

Notably, the mechanism described does not apply to germline stem cells, the most important cells for maintaining gamete production long term. One could argue that elevating damage sensitivity of early spermatogonia would be counter-productive. At low levels of radiation, the authors showed that some cyst cells die without an evidence of DNA damage. These cells would be the ideal candidates to replace damaged GSCs. Hence, it would theoretically be more beneficial to preserve undamaged cyst cells and upregulate GSC replacement, rather than killing all cyst cells regardless of their damage level. However, while an interesting subject for speculation, this question does not detract from the interest of the authors finding of a group behavior impacting damage sensitivity.

---

## [Author Response]

In reading review comments, we realize that there is a major theme that troubled reviewers: there are many examples outside male germ cells where cells are connected, in which connectivity-based increased cell death is not observed and/or such cell death does not make any (biological) sense (certain somatic cells, such as follicle cells, syncytial embryos, *C. elegans* germline, oocytes connected to nurse cells). How should the authors explain these? Many of comments revolve around this issue, and we acknowledge that our writing was misleading/confusing on this point.

First, we would like to clarify that we did not intend to claim that the only purpose of intercellular connectivity is to increase sensitivity to DNA damage, or intercellular connectivity always leads to increased DNA damage sensitivity. Based on our data as well as information from the literature (detailed in the point-by-point responses below), we speculate that it is likely the fusome instead of ‘openness (between cells)’ that specifically mediates communication between germ cells to induce their death. We acknowledge that ‘connectivity’ was too vague in this regard (can be interpreted as either communication or openness between cells), and in our revised manuscript, we clarify these points by explicitly distinguishing between ‘openness’ vs. ‘communication through the fusome’.

For example, now the Results section contains the description as following (with added data):

“It should be noted that the fusome mutants appeared to maintain ring canals, as evidenced by the intact ring shape of Pavarotti-GFP, a marker for ring canals (Minestrini, Máthé, and Glover, 2002) (Figure 3—figure supplement 2). […] Additionally, these cells did not exhibit extreme sensitivity to low doses of radiation and displayed an essentially linear dose-response (Figure 3—figure supplement 3).”

Related to this issue, the reviewers also commented on other types of connected cells (*Drosophila* embryos, *C. elegans* germline etc.). We included paragraphs on these systems in Discussion as following. Also we ensured that Introduction and Results sections do not leave the wrong impression that the increased DNA damage sensitivity is the only meaning of germ cell connectivity:

“Our present study may provide a link between two long-standing observations in germ cell biology: 1) the broad conservation of intercellular connectivity (cyst formation) of germ cells and 2) the sensitivity of the germline to DNA damage. The purpose for germ cell cyst formation outside the meroistic ovary (i.e. nursing mechanism) remained unclear. […] Therefore, we propose that the fusome functions to transmit the death-triggering signals, similar to its known role in cell cycle synchronization within the germ cell cyst (Lilly et al., 2000)(Figure 6).”

After thoroughly revising on these points, as well as adding new data (detailed below), we hope that the manuscript reads better and conveys our message more clearly.

*Reviewer #2:*

*This paper presents a fascinating, original idea regarding the origin of germ cell radiation sensitivity. The authors connect DNA damage sensitivity to the intercellular bridges that join sets of male or female germ cells into "germline cysts" for much of gamete development. Their hypothesis is strongly supported in the case of pre-meiotic 2,4,8 and especially 16-cell SG cysts. Disruption of cyst connectivity using fusome mutants or interference with the proposed damage signaling pathway using HtrA2/Omi mutants causes effects predicted by the hypothesis and the signaling mechanism. The existence of a novel group mechanism elevating damage sensitivity at least in pre-meiotic cysts is of strong interest in its own right. Shortly after these stages, during meiosis, extensive DNA damage occurs to stimulate recombination, raising the question of how the propagation of damage signals in avoided. If interconnected pre-meiotic male germ cells can transmit apoptotic signals between cells, how do interconnected female germ cells avoid damaging the oocyte during their own apoptotic process that accompanies "nurse cell dumping? "*

Thank you for the encouraging comments and interesting discussion. Indeed, it is a fascinating question to ask how some types of germ cells are interconnected but (likely) do not commit all-or-none cell death. As described above (at the beginning of our response), we did not intend to claim that any connectivity (openness) would lead to all-or-none cell death. Based on our data (and as described above), we speculate that this all-or-none death is mediated by the fusome instead of mere openings. Interestingly, in the *Drosophila* female germline, the fusome disintegrates fairly early (by region 3, soon after oocyte fate determination): thus we speculate that fusome disintegration explains how the female germline might avoid unnecessarily killing all cells within the cyst. These speculations are included in the Discussion.

Also, we now include the following data, which strengthen the idea of fusome (but not a mere openings) as a mediator of SG death:

1) Fusome mutants apparently maintains openness (ring canal) as assessed by a ring canal marker Pavarotti-GFP (Figure 3—figure supplement 2);

2) Somatic follicle cells (of the egg chamber) do not die in an all-or-none manner (Figure 3—figure supplement 3), although they are known to be connected by ring canals (McLean and Cooley).

*The argument that increased connectivity leads to increased sensitivity to DNA damage is not convincing. The authors find that there are more dying 16-cell cysts than dying gonialblasts or cysts at earlier stages. Surprisingly, they do not take into account the frequency of dying cysts at different stages relative to the abundance of cysts at these stages within the testis. 16-cell spermatocyte cysts are known to persist for ~90 hours prior to undergoing meiotic division. Since the entire period prior to meiotic division is thought to encompass ~5 days (~120 hrs), simple math would dictate that there are likely ~3x more 16-cell cysts than cysts at earlier stages of sperm development. In fact, adult testes clearly contain a preponderance of 16-cell cysts, with smaller numbers of cysts at earlier stages. Thus, the observation that there are more dying cysts at the 16-cell stage could simply be due to the fact that there are more cysts at this stage. The authors' numbers would seem to agree with this. For example, there are approximately 3x more dying 16-cell cysts at 3 and 6 hrs post-irradiation (Figure 1) than there are dying cysts at earlier stages. I strongly suggest that the authors quantify the numbers of cysts at different stages in wild-type control testes (or find references where this has been quantified), as their conclusions in this regard clearly overstep the data presented (there is no denominator) and in my view do not make sense.*

We apologize for not being clearer on this point in our original manuscript. As implied from the use of the term ‘spermatogonia’ (for which we should have been more explicit), we have limited our scoring to spermatogonial cysts only. Because spermatocytes quickly grow in size, they can be easily distinguished from spermatogonia, and any cysts with larger cytoplasm (i.e. spermatocytes) were excluded from our scoring.

The reviewers are correct in pointing out that the number of SG death/testis does not make any sense without a denominator. We have previously published that SG cysts are around ~6 per stage in a testis (Figure 2 from Yang and Yamashita 2015). Counting the number of spermatogonial cysts for each testis, within the same testis stained for SG death (Lysotracker), is not technically feasible. However, our study (Yang and Yamashita, 2015) gives us confidence that this SG cyst number/testis is fairly consistent within the genotype, and we can assume that denominator is consistent across SG stages. We have added this reasoning to the legend of Figure 1 as following:

“It should be noted that the SG death frequency was scored as ‘number of dying SG cysts at each stage per testis’. We have shown that the number of SG cysts is consistently ~5-6 cysts per stage per testis (Yang & Yamashita, 2015), justifying the use of ‘number of dying SG cysts/testis’ as a proxy for frequency of SG cyst death.”

*Moreover, support for the idea that the degree of connectivity of germ cells and communication of a DNA damage signal are not needed for increased sensitivity of developing germ cells to DNA damage is based on what is known about C. elegans germline development: although C. elegans germ cells develop within a syncytium, germ cells that are damaged by irradiation cellularize and undergo apoptosis as individual cells; DNA damage does not induce massive cell death within the large syncytium that comprises a major proportion of the C. elegans gonad. This further argues that there is no evolutionary requirement for increased connectivity in promoting germ cell death in response to DNA damage. The novelty of what the authors show for the Drosophila male germline is that connectivity leads to death of all of the cells within a cyst and that this is apparently due to sharing of a death signal among the syncytial SG cells.*

Please see our response above.

Also, regarding this comment, although the *C. elegans* germline develops in syncytium, the cytoplasm of individual germ cells is not connected due to a diffusion barrier (Cinquin et al., 2015). Therefore, it is unlikely that one damaged cell would kill all other cells (in addition to our speculation that cell death signal propagates through the fusome, as described above).

*Is the fusome itself required for the coordinated death of all SGs within a cyst or is it that the interconnectivity between cysts is what is important and that fusome loss compromises interconnectivity? How does knockdown of α-spectrin or mutation of hts influence ring canals and the ability of small molecules and proteins to move between cells within a cyst? All this may be well established or known, however, as written it was not clear to us. Please clarify this in the text. Or conduct experiment that demonstrate diffusion or lack thereof within the normal and hts mutant cysts.*

Because all-or-none SG death is compromised in fusome mutants, we believe that it is a fusome-mediated process. Ring canals (RCs) were observed to be maintained in fusome mutants, as indicated by the presence of apparently intact rings visualized by Pavarotti-GFP. We have included a figure that shows this (Figure 3—figure supplement 2). From previous literature, it was shown that the fusome both prevents ubiquitous sharing of cytoplasmic contents via RCs (Strassheim and Mahowald 1970, Cox and Spradling 2003) and is specifically necessary for facilitating sharing of certain signals between cyst members (Lilly et al., 2000). We have added these references and clarified our text throughout the manuscript to ensure that we do not mislead the readers by the term of ‘connectivity’. Our understanding is that Dr. Lynn Cooley is studying this aspect (what is shared by male RCs) and planning to submit a manuscript on this soon.

*The biggest current weakness of the manuscript is the claim that apoptosis signaling within the cyst can explain the "the extreme sensitivity" of germ cells to DNA damage. First, it is not clear that germ cell damage hyper-sensitivity has been rigorously established as what data exist derive heavily from observations of humans treated with chemotherapeutic agents. But even if germ cells are hypersensitive as a group, the authors need to explain how a mechanism that acts at relatively rare stages could strongly affect the bulk sensitivity of germ cells which are not mostly at pre-meiotic cyst stages. Consequently, the authors should more thoroughly investigate how long during meiosis and the remaining time that clusters remain interconnected does sensitivity stay elevated and all or nothing apoptosis patterns persist. If the effects are actually widespread, then this apparent difficulty would be resolved.*

This comment revolves around the fact that our original manuscript left the impression that increased DNA damage sensitivity is the only purpose for germ cell connectivity, and germ cell connectivity is the only means by which high DNA damage sensitivity is achieved. We went through the entire text as described above not to mislead the readers in such a way.

Whereas we agree that spermatogonia may be only a small fraction of germ cells within a testis, we would like to emphasize that spermatocytes are not renewable: once they become spermatocytes, they cannot amplify in number, and how sensitive or insensitive they may be to DNA damage, they will not be able to contribute to long-term fertility. Therefore, spermatogonia’s sensitivity to DNA damage would determine a significant portion of DNA damage sensitivity of the entire germline. The remaining population to be considered would be GSCs, which will be discussed below, in response to the review comments that specifically raised a question regarding GSCs.

*Similarly, if a yet unknown mechanism leads to communication of 'hypothetical death signal' between cells in cysts one would expect that other, somatic syncytia, such as the early embryos or polynucleated muscles, should be similarly sensitive. These experiments may exist in the literature or otherwise should be included.*

Please see our response above regarding this point. Now we made it clear that we speculate (with additional supportive data) that the death-promoting signal is specifically shared via the fusome, instead of mere openness. We hope that added data/discussion and the revised text is now acceptable in this regard.

*Notably, the mechanism described does not apply to germline stem cells, the most important cells for maintaining gamete production long term. One could argue that elevating damage sensitivity of early spermatogonia would be counter-productive. At low levels of radiation, the authors showed that some cyst cells die without an evidence of DNA damage. These cells would be the ideal candidates to replace damaged GSCs. Hence, it would theoretically be more beneficial to preserve undamaged cyst cells and upregulate GSC replacement, rather than killing all cyst cells regardless of their damage level. However, while an interesting subject for speculation, this question does not detract from the interest of the authors finding of a group behavior impacting damage sensitivity.*

We remain agnostic about GSCs. Dr. Erika Matunis’ group and our laboratory independently observed no (very rare) GSC death under any circumstances (including a few stress regimens such as starvation, DNA damage etc.) (Hasan et al., 2015; Yang and Yamashita, 2015). However, this does not conclude anything about the sensitivity of GSCs to DNA damage (or any other stress). It was shown in mammalian systems that stem cells simply differentiate when they are stressed (Inomata et al., 2009). Their threshold to differentiate can be more or less sensitive compared to SG death, but there is no means to compare these two distinct processes. We could speculate that GSCs might simply initiate differentiation upon DNA damage: subsequently, high sensitivity to DNA damage during spermatogonial divisions can serve as an ‘assessment method’ for their genomic integrity, and spermatogonia with any problem may die out. In this manner, spermatogonia with the highest level of genomic integrity would survive, from which GSC population can be regenerated by dedifferentiation as necessary. Of course, this scenario is a pure speculation, and might be an interesting topic for mathematical modeling in a future study. The point we would like to make here is that, our results on increased DNA damage sensitivity in spermatogonia is not necessarily counter-productive as it may seem.